# GRAPH-GUIDED CONCEPT SELECTION FOR EFFICIENT RETRIEVAL-AUGMENTED GENERATION

## ABSTRACT

Graph-based RAG constructs a knowledge graph (KG) from text chunks to enhance retrieval in Large Language Model (LLM)-based question answering. It is especially beneficial in domains such as biomedicine, law, and political science, where effective retrieval often involves multi-hop reasoning over proprietary documents. However, these methods demand numerous LLM calls to extract entities and relations from text chunks, incurring prohibitive costs at scale. Through a carefully designed ablation study, we observe that certain words (termed concepts) and their associated documents are more important. Based on this insight, we propose Graph-Guided Concept Selection (G2ConS). Its core comprises a chunk selection method and an LLM-independent concept graph. The former selects salient document chunks to reduce KG construction costs; the latter closes knowledge gaps introduced by chunk selection at zero cost. Evaluations on multiple real-world datasets show that G2ConS outperforms all baselines in construction cost, retrieval effectiveness, and answering quality.

## 1 INTRODUCTION

RAG enables large language models to access up-to-date or domain-specific information, significantly improving their question-answering (QA) performance without extra training Gao et al. (2023a;b); Fan et al. (2024). A representative approach is Text-RAG Lewis et al. (2020), which relies on document chunking and dense retrieval. However, such methods ignore inter-knowledge dependencies, leading to noticeable accuracy drops on multi-hop questions Peng et al. (2024). To address this limitation, recent Graph-based RAG (GraphRAG) techniques pre-construct a graph to capture these dependencies, substantially boosting RAG's accuracy in complex QA scenarios Procko & Ochoa (2024); Jimenez Gutierrez et al. (2024); Edge et al. (2024). In specialized domains such as law, medicine, and science, GraphRAG has been shown to significantly enhance the question-answering capabilities of large models Li et al. (2024a); Delile et al. (2024); Liang et al. (2024).

A core step in GraphRAG is establishing connections among documents and knowledge; however, the prohibitive construction cost prevents these methods from being deployed in real-world applications Abane et al. (2024); Wang et al. (2025). For example, MicroSoft-GraphRAG (MS-GraphRAG) Edge et al. (2024) processing a single 5 GB legal case Arnold & Romero (2022) is estimated to cost USD 33 k Huang et al. (2025b). The expensive cost is unacceptable for enterprise-scale knowledge retrieval systems, both for the initial build and for subsequent updates. To reduce the construction cost, recent studies constrain the graph structure to reduce the number of LLM calls: LightRAG Guo et al. (2024) adopts a key–value schema, HiRAG Huang et al. (2025a) and ArchRAG Wang et al. (2025) employ tree-like structures, where the key idea is summarizing multiple chunks with a single LLM invocation. Other efforts resort to coarser-grained graph construction to reduce LLM dependency; Raptor Sarthi et al. (2024) leverages vector models to build hierarchical clusters, and KET-RAG Huang et al. (2025b) introduces an entity–document bipartite graph. Although these approaches lower construction overhead, two major challenges remain. (1) Current methods address the cost issue by redesigning the pipeline, limiting the generalizability of their improvements. Moreover, in many applications, redesigning an existing GraphRAG system and building from scratch is prohibitively costly and impractical. (2) The restrictions on graph structure inevitably sacrifice the accuracy: LightRAG's key–value design limits retrieval depth, and top-down search strategy of HiRAG and ArchRAG struggles to find directly relevant entities.

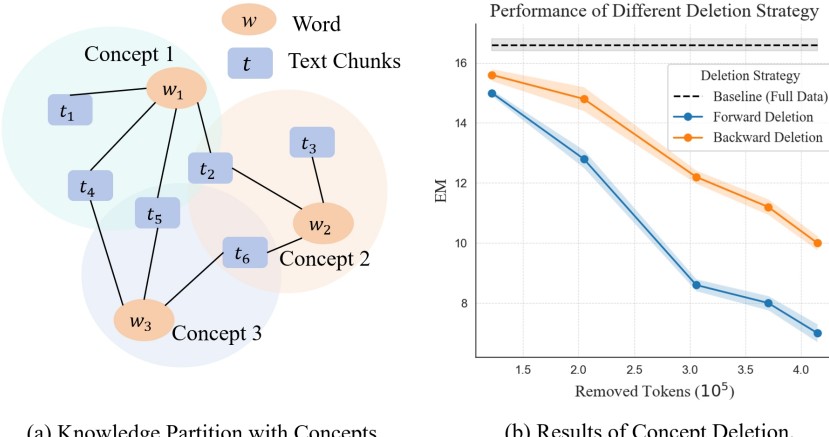

(a) Knowledge Partition with Concepts.

(b) Results of Concept Deletion.

Figure 1: The design and results of the concept deletion experiment. (a) We divide text chunks into different groups based on their associated words, referred to as concepts. (b) By deleting concepts in different orders, we find that some concepts have greater importance.

To solve the above two challenges, we propose Graph-Guided Concept Selection (G2ConS), an efficient RAG scheme that is compatible with mainstream GraphRAG approaches. Our method starts with an ablation study termed concept deletion, which investigates the importance of different knowledge to Graph RAG. As shown in Figure 1 (a), we first partition knowledge by concepts, where a concept is defined as a word along with all text chunks containing that word. After partitioning, the original knowledge document can be viewed as a composition of multiple concepts. Through concept deletion, i.e., deleting text chunks based on their group, we are able to identify the contribution of different knowledge components (indexed by concept). As a reasonable hypothesis, we claim that concepts having more connections with others are of greater importance. To illustrate this point, we first construct a concept relation graph on the MuSiQue dataset Trivedi et al. (2022): we extract concepts from text chunks using traditional keyword extraction methods Ramos et al. (2003), and then connect concepts that co-occur within the same chunks. Finally, we sort the concepts by degree, where higher-ranked concepts indicate greater connectivity to other concepts. We conduct the concept deletion with MS-GraphRAG, the results are shown in Figure 1 (b), where the x-axis denotes the number of deleted tokens and the y-axis indicates the corresponding accuracy (EM score). We evaluate two deletion strategies: forward deletion (removing concepts from highest to lowest rank) and backward deletion (removing concepts from lowest to highest rank). For comparison, we also present the performance of MS-GraphRAG with all chunks, i.e., the "baseline" in the figure. The results show that, even with equal token counts, removing high-ranked concepts still causes greater performance degradation. Based on this observation, G2ConS proposes two effective strategies. 1. For the first challenge, we introduce Core Chunk Selection, which reduces the input chunks by removing low-ranked concepts, thereby decreasing construction costs without modifying the graph construction process. 2. For the second challenge as well as knowledge gap induced by chunk selection, we propose Concept Graph Retrieval. Since the construction of the concept graph does not depend on LLM and imposes no structural constraints, it enables low-cost and effective retrieval of removed low-ranked concepts. The main contributions of the work are as follows::

- We propose G2ConS, an efficient GraphRAG scheme that retrieves from both KG and LLM-independent concept graph. G2ConS effectively balances construction cost and QA performance, outperforming state-of-the-art methods across multiple benchmarks, particularly achieving an average improvement of 31.44% across multiple metrics in MuSiQue.

- We introduce Core Chunk Selection, a general method to reduce GraphRAG construction cost with minimal accuracy loss. Combined with the Concept Graph, existing GraphRAG methods can further improve QA accuracy while reducing cost by 80%.

## 2 RELATED WORKS

**Graph-Based RAG.** Representing documents as graph structures for retrieval and question answering dates back to Min et al. (2019), who proposed constructing a graph from text chunks based on co-occurrence relations and demonstrated clear performance gains, thereby establishing the effectiveness of graph-based retrieval for QA. However, this approach ignores semantic links across documents, scattering knowledge across disconnected regions. With the rise of LLMs, many works inject them into graph construction. KG extraction methods such as GraphRAG Edge et al. (2024), HippoRAG Jimenez Gutierrez et al. (2024), LightRAG Guo et al. (2024), KAG Liang et al. (2024), FastRAG Abane et al. (2024), and GraphReader Li et al. (2024b), raising quality and QA accuracy, yet the repeated LLM calls for entity-relation extraction keep costs prohibitive at scale. Hierarchical methods like RAPTOR Sarthi et al. (2024), HiRAG Huang et al. (2025a) ArchRAG Wang et al. (2025), and MemWalker Chen et al. (2023) recursively summarize documents into layered indices; their batch processing lowers construction cost, but top-down retrieval struggles to find directly relevant knowledge, lowering retrieval accuracy. Text-level methods such as AutoKG Chen & Bertozzi (2023), PathRAG Chen et al. (2025), and KGP Wang et al. (2024) segment input texts and employ LLMs to establish inter-text or text-to-entity links. By avoiding fine-grained extraction, these approaches maintain construction costs comparable to Text-RAG Lewis et al. (2020); however, they show performance decay on multi-hop questions. In contrast, G2ConS achieves optimal performance in both construction cost and QA quality through a hybrid retrieval strategy that combines KG and concept graphs, and it is compatible with mainstream GraphRAG approaches.

**Efficient KG-based RAG.** The construction of KGs has been shown to significantly enhance the performance of RAG on multi-hop reasoning tasks Peng et al. (2024). However, the high cost of KG construction poses a major barrier to practical deployment. To address this issue, various approaches have attempted to simplify the KG construction process. For instance, LightRAG Guo et al. (2024) reduces complex node networks into key-value tables, while HiRAG Huang et al. (2025a) and ArchRAG Wang et al. (2025) construct tree-structured KGs to reduce the number of LLM calls. Nevertheless, these methods often suffer from reduced accuracy on multi-hop tasks due to their structural constraints on the KG. Another line of work explores the use of coarse-grained graphs to lower KG construction costs through lightweight named entity recognition (NER) techniques. For example, Ket-RAG Huang et al. (2025b) combines bipartite graphs with knowledge graphs to reduce overall construction overhead. $E^2$GraphRAG Zhao et al. (2025) introduces a coarse-grained entity–document chunk graph built upon LLM-generated summary trees, thereby eliminating the reliance on high-quality KGs. In contrast to these approaches, G2ConS emphasizes concept selection in graph construction and is compatible with mainstream GraphRAG methods, yielding consistent improvements in both cost efficiency and performance.

## 3 THE FRAMEWORK OF G2CONS

In this section, we first provide the problem formulation of GraphRAG. We then introduce the construction process of the concept graph, along with the implementation details of two key strategy of G2ConS: core chunk selection and dual-path retrieval.

### 3.1 PROBLEM DEFINITION

To better illustrate the problem that GraphRAG aims to address, we begin with Text-RAG Lewis et al. (2020). Given a document corpus, RAG splits the documents into text chunks according to a certain strategy. Let $\mathcal{T}$ be the set of text chunks preprocessed from documents. Text-RAG employs an embedding function $\phi(\cdot)$ to vectorize each chunk, thereby constructing an index of chunk-vector pairs: $\{(t_i, \phi(t_i))|t_i \in \mathcal{T}\}$. During retrieval, given a query $q$, Text-RAG computes the query embedding $\phi(q)$ and searches for the most relevant chunks within the chunk-vector pair index based on vector similarity. Finally, during the answer generation phase, these retrieved chunks are fed into the LLM as context, and the LLM generates the final answer in response to the query $q$. The key distinction of GraphRAG lies in its construction of a graph $\mathcal{G} = (\mathcal{V}, \mathcal{E})$ based on $\mathcal{T}$, where $\mathcal{V}$ denotes the set of nodes—each node may represent either a specific text chunk or an entity—and $\mathcal{E}$ denotes the set of edges between nodes, with edge attributes potentially including weights or textual descriptions characterizing the relationships between connected nodes. During retrieval, GraphRAG primarily follows a local search paradigm: it first identifies the nodes most relevant to the query $q$ by computing node-query similarities using the embedding function $\phi(\cdot)$, and then performs Breadth-First Search (BFS) over the graph $\mathcal{G}$ to expand the context by traversing and collecting associated

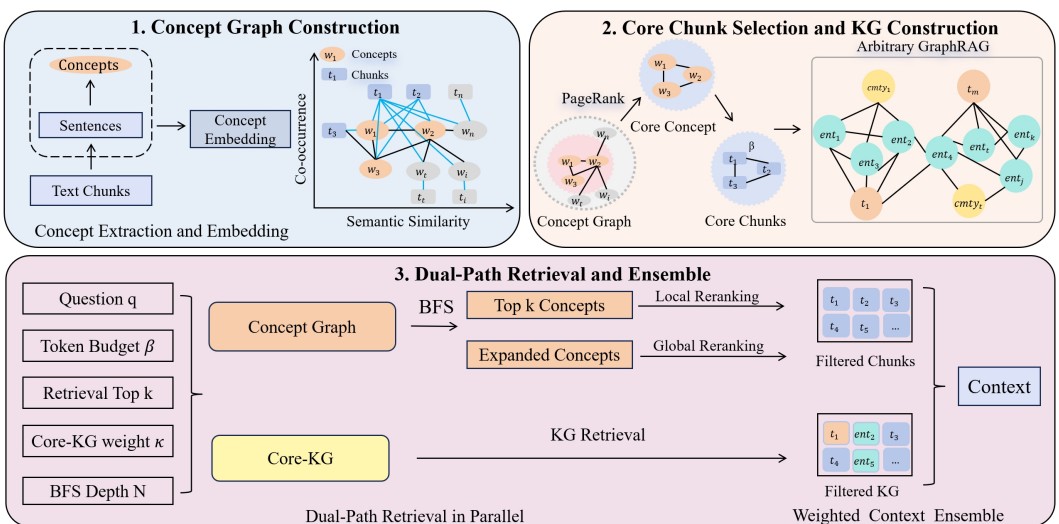

Figure 2: Overview of the Proposed G2ConS. (1) We extract concepts from text chunks and construct a concept graph based on semantic and co-occurrence relations. (2) We perform core chunk selection and build a low-cost core knowledge graph (core-KG). (3) G2ConS leverages dual-path retrieval to effectively utilize both the concept graph and the core-KG.

neighboring nodes. In the answer generation phase, GraphRAG serializes the subgraph identified by BFS and feeds it as context into the LLM to produce the final response. The objective of GraphRAG is finding $\mathcal{G}$ to maximize LLM answer accuracy Jimenez Gutierrez et al. (2024); Edge et al. (2024); in our work, we additionally impose cost constraints on the design of $\mathcal{G}$.

## 3.2 REPRESENTING CONCEPT RELATION AS A GRAPH

In Section 1, we introduced the importance of the concept graph for chunk selection as well as its construction idea. In our implementation, we further enriched the structure of the graph. As a result, the concept graph can be simultaneously leveraged for both chunk selection and retrieval.

**Concept Extraction and Vectorization.** As in Section 1, we use traditional keyword extraction methods Ramos et al. (2003) to extract concepts $\mathcal{W}$ from chunks, i.e., $\mathcal{W} = \{\text{Extract}(t_i)|t_i \in \mathcal{T}\}$. For each $w_i \in \mathcal{W}$, we construct edges between $w_i$ and its source chunk $t_i$ based on their origin. To support efficient retrieval and fuzzy entity matching, many GraphRAG approaches incorporate vectorization techniques Sarmah et al. (2024); Guo et al. (2024), and we follow the same design principle. However, by the definition of a concept — i.e., a word along with all chunks containing that word; the embedding of a single word or chunk alone cannot accurately represent the concept. On the other hand, since a chunk typically contains many tokens (often exceeding 500), its semantic representation may be contaminated by multiple concepts.

To address this, we propose constructing semantic embeddings for concepts at the sentence level. Specifically, for a concept $w_i$, we first identify all chunks connected to $w_i$, then split those chunks into sentences using a sentence splitter Loper & Bird (2002), yielding the set of all sentences containing $w_i$, denoted $\mathcal{S}_{w_i}$. We define the vector representation of $w_i$ as:

$$\text{Vector}(w_i) = \frac{1}{|\mathcal{S}_{w_i}|} \sum_{s_i \in \mathcal{S}_{w_i}} \phi(s_i), \tag{1}$$

where $\phi(\cdot)$ is the embedding function as introduced in Section 3.1. We emphasize that defining concept embeddings at the sentence level better captures the intrinsic semantics of the concept itself, and enables more accurate matching with query-relevant concepts during retrieval.

**Connect Concepts by Relevance.** According to our concept deletion experiments, co-occurrence effectively characterizes the importance of a concept. However, relying solely on co-occurrence

during retrieval, particularly when finding related concepts of the given one with BFS, tends to introduce significant noise, i.e., the spurious correlations phenomenon Calude & Longo (2017).

To mitigate this problem, edges in the concept graph are constructed by jointly considering both semantic relevance and co-occurrence in our implementation. Specifically, given concepts $w_i$ and $w_j$, an edge $e(w_i, w_j)$ exists if and only if $\text{Sim}(\text{Vector}(w_i), \text{Vector}(w_j)) \geq \theta_{sem}$ and $\text{Co}(w_i, w_j) \geq \theta_{co}$, where $\text{Sim}(\cdot, \cdot)$ denotes cosine similarity, $\text{Co}(\cdot, \cdot)$ counts the co-occurrence frequency of the two concepts across different chunks, $\text{Vector}(\cdot)$ is computed as defined in Equation 1, and $\theta_{sem}, \theta_{co}$ are constants. When $e(w_i, w_j)$ exists, we define the edge weight $r_{w_i, w_j}$ using the Dice Coefficient Li et al. (2019) to quantify the specific association strength between two concepts: $r_{w_i, w_j} = \frac{2\,\text{Co}(w_i, w_j)}{|\mathcal{T}_{w_i}| + |\mathcal{T}_{w_j}|}$, where $\mathcal{T}_{w_i}$ and $\mathcal{T}_{w_j}$ denote the sets of chunks containing $w_i$ and $w_j$, respectively. In summary, the concept graph we construct is a semantically filtered co-occurrence graph.

**Core Chunk Selection and KG Construction.** After constructing the concept graph, to better assess the global importance of concepts, we opt to rank them using PageRank Page et al. (1999) rather than local metrics like degree centrality. We note that this choice does not affect the experimental results presented in Section 1. Once concepts are ranked, we subsequently rank chunks based on which concepts they are associated with. The GraphRAG method allows selecting core chunks according to a specified proportion to construct the KG, thereby reducing the overall construction cost. In G2ConS, both the concept graph and the KG coexist. The KG is constructed by MS-GraphRAG and, since it is built exclusively from core chunks, we refer to it concisely as the core-KG.

### 3.2.1 DUAL-PATH RETRIEVAL AND ENSEMBLE

While Core Chunk Selection effectively reduces the cost, it inevitably filters out many chunks. To ensure knowledge completeness, we propose retrieving from both the concept graph and core-KG.

**Local Search on Concept Graph.** Retrieval on the concept graph largely follows the mainstream design of current GraphRAG, specifically the local search described in Section 3.1. The key difference is that we introduce a budget parameter $\beta$ to precisely control the number of context tokens retrieved. Given a question $q$, we first compute its corresponding vector $\phi(q)$, then calculate the cosine similarity between $\phi(q)$ and all concept vectors. Based on this similarity ranking, we retrieve the top-$k$ most relevant concepts, $k$ is a constant, referred to as directly associated concepts. Subsequently, using these directly associated concepts as seed nodes, we perform BFS to retrieve i-hop concepts (for i=1,...,N), termed expanded concepts. We adopt distinct context construction strategies for these two types of concepts: For directly associated concepts, we sort them by their similarity to $\phi(q)$, sequentially retrieve the corresponding chunks for each concept, and then rerank these chunks according to their similarity to $\phi(q)$ before adding them to the context. We refer to this process as local reranking. we collect all concepts discovered through BFS together with their related chunks, perform reranking over the entire set of retrieved chunks against $\phi(q)$, and then add them to the context, the process termed global reranking. Each time a chunk is added to the context, we check the accumulated token count; if it exceeds $\beta$, the entire retrieval process is terminated. We omit details of the retrieval process on the core-KG, as it follows the GraphRAG scheme in use, except that retrieved content is truncated to the token budget $\beta$. We note that retrieval over the concept graph and the core-KG proceeds in parallel to maximize retrieval efficiency.

**Weighted Context Ensemble.** After retrieval, we actually obtain two contexts, each with no more than $\beta$ tokens. However, the retrieved results differ in granularity: the concept graph consists entirely of chunks; the core-KG results include entities, relations, and chunks. Therefore, we further introduce a combination weight $\lambda \in (0, 1)$ to set preference between the two. Second, since both may retrieve overlapping chunks, we assign higher priority to overlapping chunks based on a voting strategy. Finally, we construct the final context according to the following principles: 1. ensure the total token count does not exceed $\beta$; 2. prioritize retaining overlapping chunks; 3. the concept graph occupies no more than $(1 - \lambda)\beta$ tokens, and the core-KG occupies no more than $\lambda\beta$ tokens.

## 4 EXPERIMENTS

### 4.1 EXPERIMENTAL SETUP

We evaluate our method on three widely used multi-hop QA benchmarks: Musique Trivedi et al. (2022), HotpotQA Yang et al. (2018), and 2wikimultihopqa citeho2020constructing. Following

Table 1: Key Parameters in G2ConS.

| Symbol | Description |
|---|---|
| $\theta_{sem}$ | Similarity threshold for concept graph. |
| $\theta_{co}$ | Co-occurrence threshold for concept graph. |
| $\kappa$ | Ratio of corec chunk selection. |
| $\lambda$ | Weight of core-KG in context ensemble. |
| $k$ | Number of top-$k$ items retrieved. |
| $N$ | Depth of BFS on concept graph retrieval. |

Table 2: Main Results on MuSiQue, HotpotQA, and 2WIKImultihopQA.

| Dataset Method | Musique | | | | | HotpotQA | | | | | 2wikimultihopqa | | | | |
|---|---|---|---|---|---|---|---|---|---|---|---|---|---|---|---|
| | USD | CR | EM | F1 | BERTScore | USD | CR | EM | F1 | BERTScore | USD | CR | EM | F1 | BERTScore |
| Text-RAG | 0.01 | 22.2 | 4.0 | 5.4 | 63.0 | 0.02 | 75.4 | 41.8 | 53.3 | 80.0 | 0.01 | 49.8 | 20.8 | 27.8 | 71.7 |
| MS-GraphRAG | 2.47 | 40.7 | 12.8 | 16.8 | 66.9 | 4.06 | 83.2 | 50.2 | 62.7 | 82.7 | 1.38 | 62.6 | 29.6 | 38.3 | 74.6 |
| HippoRAG | 1.99 | 45.4 | 12.8 | 17.8 | 67.0 | 3.27 | 78.3 | 46.8 | 57.6 | 83.0 | 1.78 | 68.8 | 48.0 | 54.6 | **81.0** |
| LightRAG-Local | 1.23 | 50.4 | 8.6 | 12.9 | 65.7 | 2.02 | 80.6 | 39.8 | 52.1 | 80.0 | 1.01 | 58.0 | 21.8 | 30.2 | 72.8 |
| LightRAG-Global | 1.23 | 55.2 | 9.2 | 13.1 | 65.8 | 2.02 | 84.0 | 39.4 | 52.7 | 80.3 | 1.01 | 46.8 | 6.6 | 12.1 | 66.6 |
| LightRAG-Hybrid | 1.23 | 58.2 | 9.0 | 14.9 | 66.8 | 2.02 | **88.7** | 39.4 | 53.8 | 80.7 | 1.01 | 56.6 | 19.6 | 29.1 | 72.7 |
| Fast GraphRAG | 1.47 | 31.6 | 13.4 | 18.9 | 69.2 | 2.13 | 78.4 | 46.0 | 56.3 | **84.5** | 1.12 | 62.5 | 31.0 | 35.4 | 76.1 |
| raptor | 1.10 | 43.1 | 9.2 | 15.1 | 67.0 | 1.81 | 75.0 | 44.4 | 55.5 | 81.7 | 0.62 | 48.0 | 29.6 | 36.7 | 75.0 |
| KET-RAG-Keyword | 0.03 | 60.5 | 11.6 | 17.1 | 67.2 | 0.05 | 86.7 | 48.4 | 60.9 | 82.1 | 0.01 | 64.7 | 27.2 | 34.0 | 74.1 |
| KET-RAG-Skeleton | 1.74 | 32.4 | 9.8 | 12.8 | 65.2 | 2.86 | 71.5 | 38.0 | 49.6 | 78.7 | 1.03 | 52.8 | 19.6 | 26.8 | 71.2 |
| KET-RAG | 1.77 | 50.8 | 12.2 | 17.4 | 66.9 | 2.91 | 83.4 | 45.6 | 57.9 | 81.3 | 1.04 | 64.8 | 26.0 | 33.1 | 73.6 |
| G2ConS-Concept | 0.01 | 68.4 | 15.0 | 24.1 | 69.9 | 0.02 | 84.7 | 48.6 | 62.4 | 82.7 | 0.01 | 62.7 | 35.8 | 44.4 | 77.6 |
| G2ConS-Core-KG | 1.75 | 57.2 | 13.2 | 19.9 | 68.4 | 2.88 | 75.0 | 42.4 | 54.4 | 80.0 | 1.03 | 60.8 | 31.0 | 38.5 | 75.8 |
| G2ConS | 1.76 | **71.2** | **19.8** | **29.1** | **71.7** | 2.90 | 85.5 | **51.0** | **65.1** | 83.7 | 1.04 | **70.0** | **49.6** | **55.1** | 79.9 |

prior work Jimenez Gutierrez et al. (2024); Wang et al. (2024); Huang et al. (2025b), we sample 500 QA pairs from the validation set of each dataset. For each pair, we collect all associated supporting and distractor passages to construct the external corpus $T$ for RAG. The resulting corpora contain 6,761 passages for Musique (741,285 tokens), 9,811 for HotpotQA (1,218,542 tokens), and 6,119 for 2wikimultihopqa (626,376 tokens). In terms of evaluation, we follow the common practice in multi-hop QA tasks. The retrieval quality is measured by Context Recall (CR), which assesses whether the retrieved context contains the ground-truth answer. Generation quality is evaluated by prompting an LLM to generate answers based on the retrieved context and comparing them with given answers. The evaluation metrics include **Exact Match (EM)**, which measures the proportion of predictions that exactly match the ground truth; **F1 score**, which captures partial correctness through token-level overlap; and **BERTScore**, which computes semantic similarity using BERT-based embeddings. We systematically evaluate the performance of 16 solutions, categorized into two groups: (i) Existing methods: TextRAG Lewis et al. (2020), MS-GraphRAG Edge et al. (2024), Hybrid-RAG Sarmah et al. (2024), HippoRAG Jimenez Gutierrez et al. (2024), LightRAG Guo et al. (2024) (with Local, Global, and Hybrid variants), Fast-GraphRAG AI (2024), Raptor Sarthi et al. (2024), and KET-RAG Huang et al. (2025b) (with Keyword, Skeleton, and Combine variants); (ii) Proposed methods: Concept-GraphRAG (retrieval from $G_c$ only) Core-KG-RAG (retrieval from $G_{ck}$ only, with $G_{ck}$ by default constructed using MS-GraphRAG unless otherwise specified), and G2ConS (retrieval from both $G_c$ and $G_{ck}$) and G2ConS (retrieval from both $G_c$ and $G_{ck}$). More details about baselines are in Appendix A.1. All solutions share the same inference setup: OpenAI GPT-4o-mini for generation, OpenAI text-embedding-3-small for embeddings, and Open AI tiktoken cl100k_base for tokenization. We set the maximum input chunk size to $\ell = 1200$ and the output context limit to $\beta = 12000$ across all methods. Within G2ConS, unless otherwise specified, default parameters are used with $\theta_{sim} = 0.65$, $\theta_{co} = 3$, $\kappa = 0.8$, $\lambda = 0.6$, $k = 25$, and $N = 2$. The definitions of these parameters are summarized in Table 1. We run all experiments five times and report the average values.

## 4.2 PERFORMANCE EVALUATION

We begin our empirical study by comparing **G2ConS** with representative GraphRAG baselines across three widely used multi-hop QA benchmarks: Musique, HotpotQA, and 2WikiMultihopQA. Baselines include **Text-RAG**, **MS-GraphRAG**, **HippoRAG**, **LightRAG**, **Fast-GraphRAG**, and **KET-RAG**. Table 2 reports the full results. Overall, G2ConS achieves the best balance between accuracy and efficiency, delivering strong gains in both retrieval and generation quality while significantly reducing computational cost. On Musique, G2ConS establishes the strongest overall performance. **Retrieval:** It delivers clear improvements over LightRAG-Hybrid (+22.3% CR). **Genera-**

Table 3: GraphRAG Frameworks Without vs. With G2ConS-Concept

| Dataset | Musique | | | | HotpotQA | | | |
|---|---|---|---|---|---|---|---|---|
| Method | CCR(%) | EM | F1 | BERTScore | CCR(%) | EM | F1 | BERTScore |
| lightrag(100%) | 0 | 7.2 | 12.3 | 65.8 | 0 | 37.2 | 49.8 | 79.5 |
| G2Cons-Concept+LightRAG(20%) | 80 | 10.0 | 20.3 | 69.1 | 80 | 42.0 | 57.1 | 81.6 |
| G2Cons-Concept+LightRAG(80%) | 20 | 13.2 | 21.8 | 69.2 | 20 | 43.0 | 59.1 | 82.1 |
| MS-GraphRAG(100%) | 0 | 9.2 | 15.5 | 66.0 | 0 | 45.6 | 59.4 | 82.3 |
| G2Cons Concept + MS-GraphRAG(20%) | 80 | 13.2 | 22.3 | 69.4 | 80 | 46 | 58.3 | 82.5 |
| G2Cons Concept + MS-GraphRAG(80%) | 20 | 16.8 | 25.5 | 70.6 | 20 | 49.2 | 63.9 | 83.6 |
| HippoRAG(100%) | 0 | 11.2 | 17.1 | 67.8 | 0 | 46.0 | 57.3 | 83.3 |
| G2Cons Concept + HippoRAG(20%) | 80 | 12.4 | 22.4 | 69.6 | 80 | 47.2 | 61.2 | 83.2 |
| G2Cons Concept + HippoRAG(80%) | 20 | 13.6 | 22.9 | 70.0 | 20 | 49.2 | 64.0 | 83.9 |

CCR denotes *Construction Cost Reduction*.

**tion:** G2ConS delivers substantial improvements over Fast-GraphRAG (+47.8% EM / +54.0% F1) and even larger gains over MS-GraphRAG (+54.7% EM / +73.2% F1). **Efficiency:** The lightweight variant G2ConS-Concept ranks second overall while operating at only 0.6% of G2ConS's cost, comparable to Text-RAG, highlighting the scalability of our approach. On HotpotQA, G2ConS shows dataset-dependent performance. **Generation:** It achieves state-of-the-art results, with improvements over MS-GraphRAG (+2.0% EM / +4.0% F1) and even larger gains over HippoRAG (+9.0% EM / +13.0% F1). Although the margin over MS-GraphRAG is small, G2ConS operates at about 70% of its Construction cost (a 30% reduction), underscoring a superior cost–performance trade-off. **Retrieval:** By contrast, G2ConS trails LightRAG-Hybrid, which leverages both local and global retrieval, and KET-RAG-Keyword, which boosts CR by recalling 1000 segments. However, the latter's advantage appears dataset-specific, as its CR is not consistently high across benchmarks. On 2WikiMultihopQA, G2ConS sets a new state of the art in both retrieval and generation. **Retrieval:** It slightly outperforms HippoRAG (+2% CR) and more clearly surpasses KET-RAG (+8% CR). **Generation:** G2ConS achieves small but consistent improvements over HippoRAG (+3% EM / +1% F1) and striking gains over MS-GraphRAG (+67.6% EM / +43.9% F1). **Efficiency:** Despite the close performance to HippoRAG, G2ConS operates at only 58.4% of its computational cost (a 41.6% reduction) and requires merely 5% of its graph-construction time, as shown in Figure 3(b), underscoring a significantly better efficiency profile.

## 4.3 STUDY OF COMPATIBILITY WITH MAINSTREAM GRAPHRAG

To evaluate the compatibility of G2ConS with widely used industrial approaches, we select three representative GraphRAG models—LightRAG, MS-GraphRAG, and HippoRAG—as construction backbones for the G2ConS-CoreKG Index. After constructing the G2ConS Concept Graph over the full set of text chunks, we apply PageRank to rank their importance and select the top 20% and 80% salient subsets, which are then used for knowledge graph construction. For illustration, we take LightRAG and build three indices, $G_{light100}$, $G_{light80}$, and $G_{light20}$, using 100%, 80%, and 20% of the text chunks, respectively. In the G2ConS-enhanced setting, local search results from $G_{light80}$ and $G_{light20}$ are combined with contexts retrieved from the G2ConS Concept Graph, and the merged contexts are then fed to the LLM to generate answers (with a maximum input length of $\lambda$ tokens). As shown in Table 3, integrating the G2ConS Concept Graph yields consistent improvements across all three GraphRAG methods while substantially reducing construction overhead. Even with only 20% of the text, G2ConS-enhanced models outperform their full-chunk baselines in both EM and F1, with roughly 80% lower construction cost. When 80% of the text is used, the performance gains are even larger, while costs remain about 20% lower than the baseline. These results arise because G2ConS identifies and retains highly connected concepts, thereby providing RAG methods with a compact yet semantically richer pool of candidate segments for graph construction. Dual-path retrieval ensures the LLM leverages a more informative context within the same token budget, resulting in improved accuracy and lower computational cost.

## 4.4 STUDY OF CONSTRUCTION TIME AND COST

To examine how different approaches balance efficiency and accuracy, we evaluate G2ConS against six representative RAG methods on the Musique benchmark, focusing on the trade-offs between construction cost, construction time, and performance. We adopt dataset-wide medians of cost

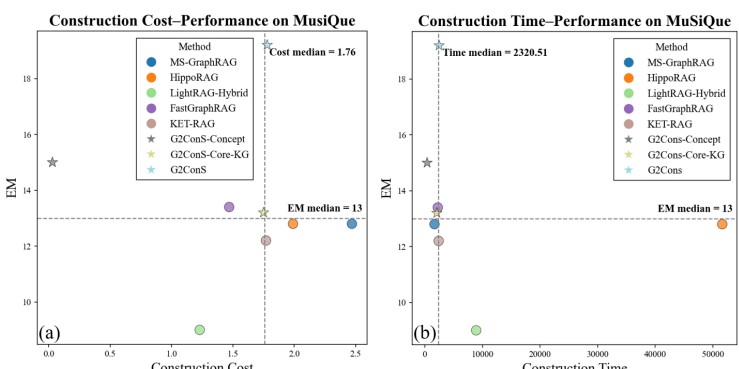

Figure 3: Construction Overhead vs. Performance on Musique.

Table 4: Ablation Study on Graph Construction and Retrieval.

| Metrics | G2ConS-Concept | A (w/o Chunk CO-occ.) | B (w/o Sem. Sim.) | C (w/o Sent. Emb.) | D (w/o LG Rerank) |
|---|---|---|---|---|---|
| EM | **13.2** | 12.0 | 12.4 | 0.4 | 6.4 |
| F1 Score | **21.8** | 20.2 | 20.5 | 1.63 | 14.6 |

and performance as reference thresholds to form a two-dimensional quadrant space, categorizing methods into four regions. As shown in Figure 3(a), **G2ConS-Concept** defines the low-cost–high-performance frontier, achieving leading accuracy with minimal overhead. **Fast-GraphRAG** also lies in this quadrant, but with higher cost and near-median performance. **G2ConS-Core-KG** sits at the intersection of the medians, serving as a balanced baseline, while **G2ConS** occupies the high-cost–high-performance region, demonstrating superior accuracy at higher construction cost. The remaining methods cluster in the low-cost–low-performance region. On the "Construction Time vs. Performance" plane (Figure 3(b)), **G2ConS-Concept** and **Core-KG** again exhibit low-cost–high-performance advantages, whereas **G2ConS** lies on the medium-cost–high-performance boundary. In contrast, **HippoRAG** and **LightRAG** require substantially longer construction times. Overall, the results highlight that the **G2ConS** family extends the efficiency–performance frontier: **G2ConS-Concept** suits efficiency-sensitive scenarios, **Core-KG** provides robust balance, and **G2ConS** further pushes the performance upper bound.

### 4.5 ABLATION STUDY

**Effect of Co-occurrence Granularity.** We compare G2ConS (edges constructed when words co-occur within a text chunk and exceed a similarity threshold) with Variant A (restricting co-occurrence to the sentence level). As shown in Table 4, on the Musique benchmark, Variant A shows a decrease of 9.1% in EM and 7.3% in F1 Score compared to G2ConS. This indicates that sentence-level windows are overly restrictive, resulting in excessively sparse graphs that miss cross-sentence dependencies, while segment-level co-occurrence better covers semantically related word pairs, yielding a more complete semantic graph. **Edge Construction with Similarity Constraints.** We further analyze the role of similarity thresholds by comparing G2ConS (chunk-level co-occurrence + similarity filtering) with Variant B (chunk-level co-occurrence without filtering). As shown in Table 4, G2ConS outperforms Variant B on the Musique benchmark. The similarity threshold effectively filters out weakly related word pairs, preventing the overly dense graphs produced by raw segment-level co-occurrence and highlighting truly meaningful semantic connections. **Effect of Vectorization Granularity.** We also examine the impact of vectorization granularity, comparing G2ConS (a concept embedding is the average of all its sentence embeddings) with Variant C (a concept embedding is the average of all its text chunk embeddings). As shown in Table 4, Variant C performs extremely poorly, with EM dropping by 97.0% and F1 Score by 92.5% relative to G2ConS. This is because segment-level averaging causes many words within the same chunk to share nearly identical representations, obscuring semantic distinctions and severely degrading retrieval quality. **Effect of Reranking Level.** There are three reranking methods: *Local*, which reranks chunks directly linked to the top-$k$ concepts; *Global*, which reranks chunks from their $N$-hop expansions; and *Naïve*, which pools both local and global chunks together without distinction. We examine the

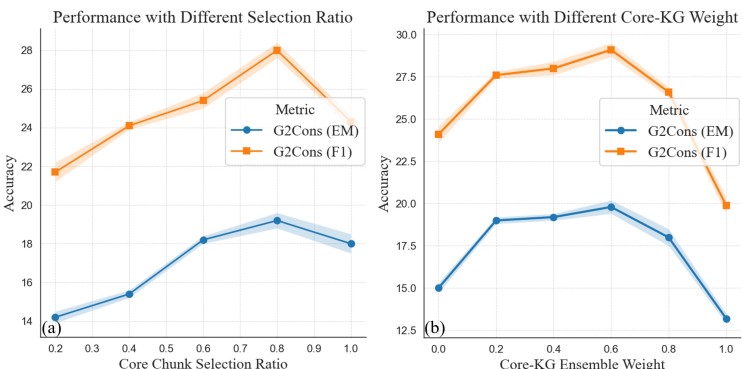

Figure 4: Answer quality by varying $\kappa$ and $\lambda$.

impact of reranking level by comparing G2ConS (Local+Global Reranking) with Variant D (Naïve Reranking). As shown in Table 4, Local+Global Reranking achieves a substantially higher EM (13.2 vs. 6.4). The advantage comes from its finer granularity: by first ranking concepts against the query and then ordering their associated chunks, it prioritizes highly relevant content. In contrast, Naïve Reranking directly sorts all chunks by query similarity, which dilutes strong signals and leads to inferior performance. **Overall Analysis.** Taken together, these ablation results demonstrate that G2ConS achieves a desirable balance between coverage and robustness in both graph construction and reranking. Segment-level co-occurrence alleviates the sparsity of sentence-level graphs by capturing cross-sentence dependencies; similarity thresholds mitigate the excessive density of raw segment-level co-occurrence by ensuring semantic reliability of edges; and sentence-level vectorization preserves word-level distinctions, avoiding the semantic collapse caused by segment-level averaging. In addition, distinguishing local and global reranking scopes allows G2ConS to leverage concept-level cues without diluting relevance, in contrast to naïve pooling. These complementary design choices enable G2ConS to consistently outperform all variants.

### 4.6 PARAMETER STUDY

We further examine the impact of two key parameters in G2ConS: the Ratio of corec chunk selection $\kappa$ and the Weight of core-KG in context ensemble $\lambda$.

**Effect of $\kappa$.** Figure 4(a) reports the performance when varying $\kappa$ with $\lambda$ fixed at 0.4. Performance improves steadily as $\kappa$ increases up to 0.6, remains near the optimum within $[0.6, 0.8]$, and then declines beyond 0.8. This pattern highlights the role of PageRank in selecting globally important chunks. **Effect of $\lambda$.** Figure 4(b) shows the effect of varying $\lambda$ with $\kappa$ fixed at 0.8. Performance grows rapidly until $\lambda = 0.2$, then increases more slowly, peaking at $\lambda = 0.6$ before declining. This indicates that optimal fusion requires balancing contributions from both G2ConS-Concept and G2ConS-Core-KG: too little $\lambda$ underuses structural knowledge, while too much diminishes semantic signals. Based on these observations, we set the default values to $\kappa = 0.8$ and $\lambda = 0.6$, which yield robust performance across benchmarks.

### 5 CONCLUSION

In this work, we propose G2ConS, an efficient RAG scheme compatible with mainstream GraphRAG. Its core idea is to mine core concepts from knowledge via co-occurrence relationships and use these core concepts to filter text chunks, thereby reducing the construction cost of GraphRAG at the data level. Experiments demonstrate that G2ConS can simultaneously optimize existing GraphRAG methods in terms of both cost and performance. Moreover, the concept graph proposed by G2ConS can independently perform RAG, achieving performance comparable to existing methods at nearly zero construction cost. In future work, we will further investigate the underlying principles of core chunk selection and conduct redundancy analysis of the concept graph to improve various aspects of G2ConS. Additionally, we will explore applying G2ConS in multimodal scenarios to build efficient RAG systems in more general settings.

**LLM Usage Clarification**: We employ the LLM solely for polishing the manuscript.

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

## A APPENDIX

### A.1 BASELINES

**Text-RAG.** Lewis et al. (2020) Text-RAG combines a pre-trained sequence-to-sequence language model with a non-parametric memory implemented as a dense vector index of external documents, such as Wikipedia. At inference time, given an input query, a pre-trained neural retriever first encodes the query and retrieves a set of relevant passages from the index. The language model then conditions on these retrieved passages to generate the output sequence. Two formulations exist: one uses the same retrieved passages for all output tokens, while the other dynamically selects different passages per token.

**MS-GraphRAG.** Edge et al. (2024) GraphRAG is a graph-based method for answering global questions over large private text corpora. It first uses a large language model (LLM) to construct an entity knowledge graph from source documents. Then, the LLM generates summaries for each community of closely related entities in the graph. At inference time, given a user question, GraphRAG retrieves relevant community summaries, generates partial responses from them, and produces a final answer by summarizing these partial responses.

**Hybrid-RAG.** Sarmah et al. (2024) HybridRAG integrates Knowledge Graph–based RAG (GraphRAG) and vector database–based RAG (VectorRAG) to improve question answering over complex financial documents such as earnings call transcripts. The method retrieves relevant context from both a vector index and a domain-specific knowledge graph, then combines these sources to generate answers.

**HippoRAG.** Jimenez Gutierrez et al. (2024) HippoRAG is a retrieval framework inspired by the hippocampal indexing theory of human long-term memory. It integrates large language models, knowledge graphs, and the Personalized PageRank algorithm to emulate the complementary roles of the neocortex and hippocampus. Given new experiences, HippoRAG constructs a knowledge graph and applies Personalized PageRank to identify relevant nodes, which guide the retrieval of supporting evidence for question answering. This process enables efficient, single-step retrieval without iterative prompting.

**LightRAG.** Guo et al. (2024) LightRAG enhances Retrieval-Augmented Generation by incorporating graph structures into text indexing and retrieval. It employs a dual-level retrieval system that operates over both low-level textual units and high-level graph-based knowledge representations. During retrieval, the framework leverages vector embeddings alongside graph connectivity to efficiently identify relevant entities and their relationships. An incremental update algorithm dynamically integrates new data into the graph, ensuring the index remains current without full reprocessing.

**FastGraphRAG.** AI (2024) FastGraphRAG is a retrieval augmented generation framework that accelerates knowledge intensive reasoning by leveraging lightweight graph structures for efficient indexing and retrieval. It constructs a compact entity relationship graph from input documents and applies fast approximate graph traversal such as personalized PageRank or random walks to identify relevant context in a single retrieval step. By integrating graph based semantic connectivity with dense vector representations, FastGraphRAG enables rapid access to both local and multi hop evidence while minimizing computational overhead.

**Raptor.** Sarthi et al. (2024) RAPTOR is a retrieval augmented language model that constructs a hierarchical tree of text summaries through recursive embedding, clustering, and summarization of document chunks. Starting from fine grained segments at the leaves, the method progressively groups and summarizes content to form higher level abstractions toward the root. During infer-

ence, RAPTOR retrieves relevant nodes across multiple levels of this tree, enabling integration of information at varying granularities and supporting holistic understanding of long documents.

**KET-RAG.** Huang et al. (2025b) KET-RAG is a multi-granular indexing framework for Graph-RAG that balances retrieval quality and indexing efficiency. It first selects a small set of key text chunks and uses a large language model to extract entities and relationships, forming a compact knowledge graph skeleton. For the remaining documents, it constructs a lightweight text keyword bipartite graph instead of a full triplet based knowledge graph. During retrieval, KET-RAG performs local search on the skeleton while simulating a similar traversal on the bipartite graph to enrich context.

