# OpenReview forum: "Graph-Guided Concept Selection for Efficient Retrieval-Augmented Generation"
_ICLR.cc/2026/Conference — Submitted to ICLR 2026_

### Official Review · Reviewer_mFAp · 2025-10-30

**Soundness:** 2
**Presentation:** 3
**Contribution:** 2
**Rating:** 2
**Confidence:** 4

**Summary:**

This paper proposes a approach for improving retrieval-augmented generation (RAG) systems by using graph-guided concept selection mechanisms. The method aims to enhance the efficiency and accuracy of information retrieval in RAG frameworks by leveraging graph structures to guide the selection of relevant concepts.

**Strengths:**

S1: The paper provides comprehensive comparisons with many other mainstream GraphRAG approaches, demonstrating a thorough understanding of the current landscape.

S2: The presentation is relatively clear and well-structured, making the methodology accessible to readers.

**Weaknesses:**

W1: The rationale for using BFS algorithm for retrieving top-k concepts in the Dual-Path recall is unclear. Based on my understanding, a concept neighborhood graph should be constructed, and greedy beam search should be employed for retrieval.

W2: The paper lacks a REPRODUCIBILITY STATEMENT as recommended by the ICLR Submission Guidelines. The entire methodological framework is only provided through flowcharts without specific algorithms (not even pseudocode). Unless I missed it, I did not see any supporting materials for the reproducibility of your paper.

W3: As a methodological contribution, your approach lacks fundamental theoretical support and theoretical analysis. This makes your method's contribution solely dependent on experimental results, which I cannot reproduce.

W4: The contribution is insufficient and lacks innovation. There is no essential difference from many other methods that combine text similarity and knowledge graph construction; it is merely an engineering implementation/optimization.

**Questions:**

Q1: After constructing edges between concepts through semantic relevance, do you provide attributes? If you do provide attributes, how are these attributes obtained, and how do they compare to LLM-based methods? If you don't provide attributes, does this mean that, compared to traditional methods using LLMs to extract associations, this is actually just a trade-off/compromise in this aspect?

Q2: Referring to W2, can you provide more reproducible details?

Q3: The content is relatively sparse. Could you provide more theoretical analysis, experimental analysis, and methodological explanations?

**Details Of Ethics Concerns:**

No Ethics Statement was found in the paper.

---

### Official Review · Reviewer_6HTH · 2025-10-31

**Soundness:** 2
**Presentation:** 2
**Contribution:** 3
**Rating:** 4
**Confidence:** 4

**Summary:**

This paper introduces G2ConS (Graph-Guided Concept Selection), an efficient variant of Graph-based Retrieval-Augmented Generation (GraphRAG) for large language model (LLM) question answering. Motivated by the high computational and monetary cost of constructing knowledge graphs (KGs) through extensive LLM calls in existing GraphRAG pipelines, the authors present an ablation study showing that certain "concepts" (defined as words and their associated documents) are markedly more important for QA performance. G2ConS leverages this insight via two main strategies: Core Chunk Selection (filtering KG construction to only chunks containing important concepts, as ranked via a PageRank on a lightweight, LLM-independent concept graph) and Concept Graph Retrieval (compensating for removed knowledge by using the concept graph for additional retrieval at negligible cost). They implement a dual-path retrieval scheme, merging knowledge from both the core-KG and concept graph, and provide results on multi-hop QA datasets, demonstrating cost savings and improved or maintained QA performance compared to prior GraphRAG and related baselines.

**Strengths:**

- The motivation for reducing reliance on LLM calls in graph construction is strong and addresses a clear inefficiency in existing GraphRAG approaches.
- The idea of constructing concept graphs based on word and chunks co-occurrences, rather than using costly LLM-based chunk selection or summarization, is novel and interesting.

**Weaknesses:**

- The generalization of the core component, Core Chunk Selection, is questionable. This component constructs relations between words and text chunks and then filters information simply based on ranking. This may lead to information loss, and there is no quantitative metric for how much information needs to be removed. The process lacks a convincing discussion of its generalizability.
- The construction and effectiveness of the concept graph in G2ConS may be sensitive to parameter choices (e.g., similarity threshold θsim = 0.65 and co-occurrence threshold θco = 3). The paper does not provide ablation studies for these hyperparameters, raising concerns about robustness and generalizability.
- The motivation for introducing the core-KG into the G2ConS framework is not clearly explained. Although experiments show that combining core-KG and concept graph retrieval achieves better results, the necessity of including core-KG is not justified. Additionally, introducing an extra retrieval pool increases both retrieval and knowledge merging costs.
- There is a repeated sentence on line 309: "(ii) Proposed methods: Concept-GraphRAG (retrieval from Gc only), Core-KG-RAG (retrieval from Gck only, with Gck by default constructed using MS-GraphRAG unless otherwise specified), and G2ConS (retrieval from both Gc and Gck ) and G2ConS (retrieval from both Gc and Gck )."

**Questions:**

Please refer to the weaknesses section

---

### Official Review · Reviewer_3H3Y · 2025-10-31

**Soundness:** 1
**Presentation:** 1
**Contribution:** 1
**Rating:** 2
**Confidence:** 5

**Summary:**

While the paper aims to tackle the relevant problem of high KG construction costs in GraphRAG by constructing a mindmap like concept graph, the proposed solution, G2ConS, lacks the novelty and technical depth expected for ICLR. The core idea of concept selection is straightforward, the methodology relies heavily on simplistic and conventional methods, also many unlearnable hyperparameters, and the experimental evaluation is insufficient to support the claims of significant advancement.

What shocks me is that authors also construct a traditional KG using MS-GraphRAG as another retrieval path separately, then what is the meaning of the designed concept graph (let alone there is no coupling between them)...since MS-GraphRAG is widely verified as an early pioneering work but performs the worst and costs the most and conflicts with their motivation.

**Strengths:**

- Authors had a simple but clever idea: ​​not all information is equally important.​​ They ran experiments where they selectively deleted concepts and their related text chunks from the knowledge base.
- Considering the concept graph itself, it is somehow a contribution to the situation, but what a pity, authors make it in a wrong way.

**Weaknesses:**

- The premise that some concepts are more important than others is intuitive and not a novel finding. The proposed method only relies on traditional NLP methods into an optimization "plugin" for existing systems. This work feels more like an engineering improvement than a fundamental algorithmic or theoretical contribution that pushes the boundaries of the field.
- What a pity that this paper makes it in a wrong way that it combines a core-kg from MS-GraphRAG as another retrieval path separately. This exactly conflicts with their motivation to save costs! then what is the meaning of the designed concept graph (let alone there is no coupling between them)...since MS-GraphRAG is widely verified as an early pioneering work but performs the worst and costs the most...
- The concept graph heavily relies on many hyperparameters in a training-free way. This makes this engineering trick much less scalable.
- The comparisons lack depth against a broader range of SOTA GraphRAG methods. It remains unclear if the performance gains are due to the graph structure or simply from the dual-retrieval mechanism, let alone the defining and detecting concept graph is over-simplified.

**Questions:**

- How were these hyperparameters determined? Were they tuned on the validation set of each benchmark?
- Why is the Core-KG path combined and necessary? If the Concept Graph is effective at retrieving relevant information at near-zero cost, what specific deficiencies does it have that the Core-KG path compensates for? Is it a lack of relational reasoning, lower-quality textual context, or something else? also this conflicts with your motivation since MS-GraphRAG is one of the most expensive GraphRAG methods.

---

### Official Review · Reviewer_KsSd · 2025-11-01

**Soundness:** 2
**Presentation:** 2
**Contribution:** 1
**Rating:** 2
**Confidence:** 4

**Summary:**

This paper aims to address the high costs in GraphRAG (frequent calls to LLM for entity and relation extraction).  Specifically, a conceptual framework known as Graph-Guided Concept Selection (G2ConS) is developed. The key intuition behind this is “Core Chunk Selection” in G2ConS. Firstly, a “Concept Graph” without relying on an LLM is established to estimate weights for all chunks in a text. Then expensive KG building is conducted for only selected “core” chunks. Thus, a substantial cost is saved. In order to compensate for possible knowledge deficits caused by chunking centered around core chunks, the framework follows a strategy of Dual-Path Retrieval. The strategy jointly searches the core KG and the Concept Graph and combines their search results through weighted fusion. Experiments on multiple multi-hop QA benchmarks (e.g., MuSiQue) demonstrate that G2ConS outperforms baselines in both cost efficiency and answer quality. Moreover, it can serve as a plug-in module, reducing the cost of existing GraphRAG methods by approximately 80% while improving their accuracy.

**Strengths:**

1. Graph-Guided Concept Selection (G2ConS) will ensure an efficient and economical process for constructing a graph.

2. The idea of building a Concept Graph that is independent of LLM and uses its structure for linking text segments is somewhat novel. The conceptualization of the Dual-Path Retrieval mechanism is insightful.

3. Extensive experiments on several QA benchmarks demonstrate the effectiveness of the proposed method.

**Weaknesses:**

1. The overall motivation and implementation of this work are highly similar to KET-RAG, which also employs a KG skeleton concept and a dual-path retrieval mechanism to achieve low-cost graph construction. Therefore, this paper appears to be an incremental extension of KET-RAG. I think its novelty seems insufficient.

2. The main experiments are conducted only on the GPT-4o-mini model, without evaluation on open-source models such as the LLaMA or Qwen series. This raises concerns about whether the proposed method can remain effective when applied to models with weaker reasoning capabilities.

3. The ablation study setup is quite simple, as it is conducted only on the MuSiQue dataset. This limited scope is insufficient to fully demonstrate the design advantages of the proposed method.

4. The Parameter Study section (Section 4.6) is presented too briefly. It is unclear which dataset was used for the evaluation in Figure 4. Moreover, the results suggest that the method may be sensitive to hyperparameter selections. Would the optimal hyperparameters differ across different datasets? This point should be further clarified and discussed.

5. The paper lacks certain important implementation details that should be included in the appendix for better clarity. For example, how is the Context Recall (CR) metric specifically calculated? How are the hyperparameters of different baselines configured? Were there any rules or strategies applied in question sampling, or was it purely random? These aspects should be clearly described in the appendix to improve the paper’s transparency and reproducibility.

6. No shared code, and this paper is not clear enough to reproduce the results.

**Questions:**

See above.

---

### Meta-Review · Area_Chair_x2Ez · 2026-01-05

**Summary:**

The motivation of this paper is seriously concerned by the reviewers. Besides, the algorithm design also contains critical drawbacks.

**Reviewer Concerns:**

The authors did not provide the rebuttal, so the original concerns remain.

**Reviewer Scores:**

The reviewers are unlikely to change their scores.

---

### Decision · Program_Chairs · 2026-01-26

Reject